# Comparative UAV Noise-Impact Assessments through Survey and Noise Measurements

**DOI:** 10.3390/ijerph18126202

**Published:** 2021-06-08

**Authors:** Jurica Ivošević, Emir Ganić, Antonio Petošić, Tomislav Radišić

**Affiliations:** 1Faculty of Transport and Traffic Sciences, University of Zagreb, 4 Vukelićeva Street, 10000 Zagreb, Croatia; tradisic@fpz.unizg.hr; 2Faculty of Transport and Traffic Engineering, University of Belgrade, 305 Vojvode Stepe Street, 11000 Belgrade, Serbia; e.ganic@sf.bg.ac.rs; 3Faculty of Electrical Engineering and Computing, University of Zagreb, 3 Unska Street, 10000 Zagreb, Croatia; antonio.petosic@fer.hr

**Keywords:** drones, noise sources, public health, noise annoyance, public acceptance, UAV, noise measurement, survey

## Abstract

Possibilities to use unmanned aerial vehicles (UAVs) are rapidly growing. With the development of battery technologies, communication, navigation, surveillance, and autonomous systems in general, many UAVs are expected to operate at relatively low altitudes. Thus, the problem of UAV noise impact on human health and well-being will be more pronounced. In this paper, we conducted noise measurements of two UAVs of different performance (quadrotor and hexarotor) in flying up and down, hovering, and overflight procedures. Respondents of good hearing who were confirmed by audiogram measurement and had participated in the survey during UAV noise measurement gave their subjective assessments on the UAV noise perception. UAV noise measurements and subjective respondents’ assessments were analysed and related. UAV noise analysis showed that the parameters measured at the same measurement point for the hexarotor were higher than those for the quadrotor in flying up and down and flying-over procedures. Low frequency noise was present in the noise spectrum of both drones. Participants were able to distinguish between the noise of UAVs and had a generally more negative experience with the hexarotor. Regardless of the noise perception, more than 80% of the respondents believe there are more pros than cons for UAV introduction into everyday life.

## 1. Introduction

Potential uses of unmanned aerial vehicles (UAVs) are versatile. Not only do they find application in recreational activities, such as photography, racing, and sports, but UAVs are also useful for non-recreational uses that include law enforcement, emergency response, media coverage, surveying, and utility inspection [1]. Use of UAVs could, in some cases, increase efficiency by reducing the time and personnel needed to complete specific tasks without putting a human life at risk [1], such as during pavement and facility inspections, wildlife hazard management, search and rescue operations [2], security management, and in many other applications [3]. UAVs also show a significant potential to facilitate deliveries [4,5], especially of critical and discretionary items, such as biological material [6], blood products [7], vaccines [8], medical devices [9], and medicines for medical home care during pandemic, such as COVID-19 [10].

Despite all the benefits, introduction of UAVs and their integration into day-to-day air traffic management operations and planning is associated with many risks [11] and represents a major challenge for airports, air navigation service providers, and the diverse stakeholders involved. Inclusion of UAV operations imposes various safety, economic, operational, regulatory, community, infrastructure, and environmental issues to the current concept of operations [1].

Undoubtedly, the increase in UAV activities will lead to additional adverse effects on the environment, particularly due to noise issue, which is recognised as one of the major concerns regarding the use of drones in urban environments [12]. New approach and departure procedures, introduced by the new UAV traffic, will in turn result in new aviation noise exposure, especially in urban areas [13]. With the increased use of UAVs, the current number of noise complaints is expected to grow [14].

Even though, in many research papers, drones are described as a more environmentally friendly technology (especially in terms of CO_2_ reduction) for both logistics and passenger transportation [15,16,17,18,19], introduction of new modes of transport, such as person-carrying drones, can cause increased annoyance compared with other transportation modes [20,21]. Due to the higher loudness, sharpness, and tonality of the quadrotor, the calculated psychoacoustic annoyance proved to be higher than for the aircraft and road vehicles tested [22]. In addition to the analysis of UAVs’ noise levels, the additional cognitive load induced by these new vehicles should also be researched as a potential noise impact [23].

In general, hearing impairment is the most recognized consequence of long-term exposure to excessive noise levels. In addition, chronic exposure to large-pressure amplitude and low-frequency noise produced by drones (≥90 dB SPL, ≤500 Hz) can impact the human body, causing irreversible organic damage to the organism, such as neurological disturbances, respiratory disorders, and cardiovascular problems [24]. It was found that the solidity of fused cochlear cilia, both amongst themselves as well as to the upper tectorial membrane, causes discomfort (i.e., annoyance) when the basal membrane moves in response to the presence of an acoustical stimulus [25].

A range of UAV noise-related research included acoustic measurements in laboratories and an anechoic chamber, while some research also involved field testing. An acoustic-signature profile generation of a small, multi-rotor, unmanned aircraft system established a relationship between distance, altitude, and sound pressure level [14]. The measurement of UAV noise using a phased array microphone system at Jingmen airport in the Hubei province of China showed that data acquired by a phased array can be used for identification of source locations as well as spectral analysis of noise sources with appropriate treatments [26]. In several research studies, noise measurements were conducted with the aim to detect the presence of small UAVs inside a closed environment by distinguishing the typical tonal component of UAV noise [27] or by using its sound power level [28]. In addition, Iannace et al. [29] used acoustic measurements to train a model based on logistic regression for automatic detection of unmanned aerial vehicle, while Djurek et al. [30] proposed an algorithm for detection of a quadcopter by passing the recorded audio signal through a nonlinear function, which could identify the differences in the signal’s spectral components.

There are several studies that compare outdoor noise measurements to the ones performed inside anechoic chambers. One such comparison was conducted for four commercially available eVTOL propellers in an anechoic chamber and drone park at Virginia Tech [31]. The results showed that operating the same propellers in an anechoic chamber and in an open environment produced consistent tonal levels at the blade-passing frequency (BPF) and its harmonics. Another research stressed that due to the nature of testing in a closed facility, such as anechoic chambers, the effects of flow recirculation led to an increase in noise, which, for a similar rotor in hovering flight, did not occur outdoors [32].

Additionally, perception of UAV noise might be different when running the listening experiments outdoors in contrast to the anechoic chamber. Some of the limitations of outdoor testing are connected to the effects of ground reflections, ambient noise, as well as meteorological factors. Even though the absence of all these factors during anechoic chamber experiments can lead to a more pronounced perception of UAV noise, their presence outdoors makes the listening experiments more realistic.

Several research efforts went beyond noise measurement and proposed noise reduction measures for UAV operations. One study explored a passive noise-reduction approach based on micro-perforated panel (MPP) absorbers together with sound-proof materials that, as a result, can achieve significant reduction in the motor’s overall noise level [33]. However, more common are research contributions focused on reducing the noise from propellers. One of the solutions to mitigate noise proposed an experimental investigation of aero-acoustic properties of a propeller [34], while another examined a possible noise-reduction technique for a small-size propeller that is achieved through boundary-layer tripping by a series of static and wind-tunnel tests [35]. One of the methods for reducing propeller noise is to surround the engine rotors with ducts (fabricated using 3D printing) designed to absorb their acoustic emissions [36].

While public acceptance of UAV operations in urban areas will most certainly depend on many factors, such as safety issues and associated risk perceptions [37,38,39,40], it goes without saying that UAV acoustic characteristics and the community’s response thereto (which is currently uncertain) will also play a crucial role. Therefore, more research on human perception and reactions to this novel noise source is needed.

Researchers at NASA Langley conducted a psychoacoustic test on 38 subjects to investigate their levels of annoyance to 103 pre-recorded, non-unique sounds of different vehicles, focusing on small, unmanned aerial systems (sUAS) [20]. The initial analysis indicated that there may be a systematic difference between the annoyance response generated by the noise of the sUAS and the road vehicles included in this study. Another experiment involved a series of listening tests made of audio-visual signals simulating a drone hover in seven urban scenes presented to 30 healthy participants who were asked to report their perceived loudness, annoyance, and pleasantness for each one [21]. One of the conclusions indicated that road traffic could mask drone noise, suggesting that UAV corridors might be defined along road infrastructure to alleviate the noise impact on residents.

Since most of the listening experiments were conducted indoors with pre-recorded audio-visual scenarios, more outdoor testing of UAV operations involving human subjects is needed to provide a more realistic experience to participants of such experiments when assessing the impact of UAV noise on the population.

In this paper, noise measurements of two UAVs of different performance (quadrotor and hexarotor) in take-off, landing, hovering, and overflight are reported. Respondents of good hearing, who were confirmed by audiogram measurement and had participated in the survey during UAVs noise measurement, gave their subjective assessments on the UAV noise perception. The UAV noise measurements and subjective respondents’ assessments were analysed.

The structure of the paper is as follows. Section 2 provides a brief description of the methodology behind the noise measurements and survey conducted. Results and discussion are presented in Section 3 along with the UAV noise analysis, respondents’ data, and relation between measured noise and subjective assessment. Finally, some conclusions, remarks, and ideas for future work are presented in Section 4.

## 2. Methodology

### 2.1. Measurement Location and Meteorological Data

During selection of the measurement location, care was taken to comply with the requirements of Section 4.2 of ISO 20906 [41] and Appendix C of ISO 1996-2 [42]. These standards describe selection of a sound measurement site as a two-stage process. In the first stage, a measurement site is selected in general, based on the set measurement objectives. In the second stage, a specific measurement site within the general area is selected based on practical and other considerations, such as interference from other sound sources, reflections from terrain and buildings, measurement uncertainty, etc.

Since the measurement objective of this research is to obtain accurate sound information from proprietary noise sources—drones—the general area was not determined by a specific location. In view of practical and other requirements, such as the need for sufficient space where the operation of drones would be possible, the Borongaj Campus of the University of Zagreb was chosen as the measurement location.

University Campus Borongaj covers an area of 92.8 hectares in the eastern part of the city of Zagreb and is characterised by uninhabited, large, open, natural areas and rare development with mostly low buildings. Precisely for the above reasons, an open meadow in the central part of the campus was chosen as the measurement point. The area of the part of the meadow where the measurements were made covers a little less than 500 m^2^ and is located between latitudes 45°48′49.3″ N and 45°48′48″ N and longitudes 16°02′35.5″ E and 16°02′36.4″ E.

To ensure minimal uncertainty in the sound-level measurements, all acoustically relevant reflective surfaces except the ground were at least 10 m away from the microphone located in the free field, which is in accordance with the ISO 20906 standard. On the north and east sides of the measurement area, there is an open meadow stretching more than 300 m. On the south side of the measurement area, there is a local road with very infrequent traffic at 7 m, and the nearest single-storey building is a library, at 25 m. On the west side of the measurement area, at 30 m, there is a student restaurant. The measurement polygon in relation to the neighbouring buildings, the road, as well as the open meadow is shown in Figure 1.

The UAV noise measurements and respondents’ assessments were conducted separately on two different days. The UAV noise measurements were conducted on Wednesday, 11 November 2020 from 10:15 a.m. to 1:15 p.m. The subjective respondents’ assessments were conducted on Thursday, 19 November 2020 from 9:25 a.m. to 11:25 a.m.

Meteorological conditions (temperature, humidity, wind velocity magnitude perpendicular and parallel to the microphone membrane surface) were measured using wireless anemometer sensors. Meteorological data were measured parallel with the noise measurements, and the results are shown in Table 1.

According to ISO 1996-2:2017 standard (Chapter 5.1.1), the influence of the wind-speed magnitude on aerodynamic noise generated by microphone was negligible.

Detailed meteorological conditions were not monitored at 10 m height (wind speed and direction at UAV flying altitude) to determine meteorological windows according to ISO 1996-2:2017, because the maximum distance from the UAVs to the receiving point was shorter than 50 m. Consequently, the influence of meteorological conditions on the measured sound pressure levels can be neglected [41,42,43,44,45,46]. Local wind turbulence may have had some effects on the drone stability, which was visible in the same types of operations. Therefore, the same types of operations were repeated several times and averaged according to the procedures described in ISO 1996-2:2017.

The subjects of the noise measurements were two multirotors: a small commercial quadrotor and a custom-built hexarotor. Details of the UAVs are presented in Table 2.

The maximum flying time for each UAV was carefully considered, as this is one of the most critical criteria for the success of the practical part of the research conducted. Since the UAV operations were performed both during noise measurements and assessments by respondents, each day of the experiment, every drone was flying for about 20 min, which coincided with the duration of one battery. Flight time of each drone depends on the capacity of its battery and its payload. For the drones used in this study, flight time was between 20 and 40 min. Furthermore, at least two reserve batteries for each drone were in place to ensure smooth running of the experiment.

### 2.2. Measurement Equipment

Two measurement setups (Class 1) were used for UAV noise measurements. The first consisted of a measuring microphone, an external acquisition card, and software to display and analyse the sound measurement data. The second measurement setup consisted of a precise, handheld sound-analyser device and was used to control measurement results at the same measurement location.

Both measurement setups comply with the following standards applicable to sound level meters, 1/1 octave and 1/3 octave filters: IEC 61672-1:2002 Class 1, IEC 60651 Class 1, IEC 60804 Class 1, IEC 61260 Class 1, ANSI S1.4-1983 (R2001) with amendment S1.4A-1985 Class 1, ANSI S1.43-1997 (R2002) Class 1, ANSI S1.11-2004 Class 1. The above-mentioned equipment has a wide range of applications and features, only some of which were used in these measurements.

In addition, hearing ability of respondents was measured using a Class IIa medical audiometer according to Annex IX of Directive MDD 93/42/EEC, as amended by 2007/47/EC. Its main feature is diagnostic pure-tone audiometry: air or bone conduction, the first of which was used in these measurements.

### 2.3. Respondents’ Data, Audiometry, and Subjective Assessment of Noise

Thirty-one respondents of good hearing were enrolled in the study. The study protocol was approved by the Faculty’s Ethics Committee, and participants provided their informed consent in written.

The respondents were male and female, aged 21 to 61, with an average age of 31 years. All participants were screened for hearing impairment by determining the average hearing threshold in each ear at typical octave frequencies (0.5 kHz, 1 kHz, 2 kHz, and 4 kHz). This audiometry was performed in a climate-controlled room at 21 °C, relative air humidity of 50–60%, and negligible background noise. The average hearing level of all participants was between 0.55 dB and 17.25 dB, well below the World Health Organization’s threshold for hearing impairment of 25 dB.

After the audiometry, the content of the survey questionnaire and the research procedure were explained to the respondents. The survey consisted of 4 pages and a total of 6 questions. The first question task was to rate their agreement with the statements by considering the experience of quadrotor noise exposure by marking anywhere on the scale from −5 to +5. Continuous scale was defined to avoid integer discrete responses, where −5 means “I completely disagree”, 0 means “I have no attitude/I don’t care”, and +5 means “I completely agree”. The question consisted of 8 statements listed below:S1.Being exposed to 1 h of UAV noise like this one would have a negative impact on my mood.S2.Daily UAV operations like this one would make me afraid for my safety due to possibility of UAV crash.S3.Daily UAV operations like this one would have a negative impact on my quality of life.S4.Being exposed to 1 h of UAV noise like this one would have a negative impact on my focus even when performing simple mental tasks.S5.I think that being exposed to 8 h of UAV noise like this one would cause a permanent hearing loss.S6.Even short-term exposure to this type of noise causes ringing in my ears.S7.I think that UAV noise like this one makes communication difficult.S8.Based on the experience in this experiment, I absolutely do not approve of UAV operations in my vicinity.

The second question had the same task and statements as the first one and was related to the hexarotor noise. 

In the third question, the respondents were asked to provide their general data, such as date of birth and gender. 

In the fourth question, the respondents were instructed to choose the two most important reasons for preventing the introduction of drone operations into everyday life. The offered reasons were as follows:The noise of drones would negatively affect my mood.Injuries are possible due to possible accidents caused by drone operations.Drone noise would reduce my quality of life.I’m afraid drone noise would permanently damage people’s hearing.Due to the UAV noise, it would be difficult for me to communicate in general.

In the fifth question, the respondents were asked to choose two most important reasons for introducing drone operations into everyday life. The offered reasons were as follows:I support every technology with the goal of improving the quality of life.I support the use of green technologies, especially if they reduce environmental pollution.I want the possibility of delivering goods (e.g., food or shipments) by drone directly to the home address.I want to reduce the time of delivery of goods to the home address.I want the opportunity to use a drone taxi as soon as possible.

The sixth question task was to rate their agreement or disagreement with the statement in the same way as for the first two questions, and the statement was: “Given the overall experience in this research, I believe that the benefits of introducing drone operations into everyday life outweigh their negative impact”.

Assessments by respondents were conducted in one day. The first part of the experiment involved up and down flying procedures with the quadrotor, during which the respondents were placed in a semicircle at equal distances from the drone. Secondly, the flying-over procedure was conducted with the respondents placed in a row (trellis) at equal distances from the drone. After the quadrotor flying procedures were completed, the respondents answered the first question in the survey. The same procedure was repeated with the hexarotor, after which the respondents answered the second question in the survey. After the flying procedures of both drones were performed, the respondents answered the remaining four questions in the survey.

### 2.4. Noise Measurements

The measurement situation with a free-field microphone located at a height of 1.7 m is shown in Figure 2. Minimum distance from the closest flying point was 4.5 m while ascending and descending and 8.3 m while flying over the measurement position. The ground covering was short-cut grass.

The sound pressure signals were recorded at a sampling frequency of 50 kHz, and the measurement results are shown as the A-weighted equivalent sound pressure levels (with FAST time integration constant and 1 s integration interval). The average speed during the ascent and descent to 10 m was 5 m/s, while the average speed during flying over procedure at 10 m was 3 m/s.

The situations with direct and reflected ray paths for two drone positions during the up- and down-flying procedure and for flying-over-microphone-position procedure are also shown in Figure 2. It is evident that distance from the microphones changed in time, and maximum and equivalent sound pressure levels were measured for all considered flying operations.

Due to the low average speed during the ascent and the descent, when the UAV was passing through the microphone (observers), the Doppler shifting in the frequency can be neglected. This claim is supported by the fact that the direction toward the microphone (observer) was perpendicular to the moving directions, especially in the moments when the instantaneous sound pressure level reached maximum value.

For the flying-over procedure, frequency shifting appeared when the drone was flying toward and from the measurement position. Nevertheless, due to the relatively low flying speed (<10 m/s) compared to the speed of sound (c = 343 m/s) multiplied with basic harmonic of noise spectrum (f0) at considered atmospheric conditions, the shifting of the frequency components (Δf = Δv/(c × f0) can also be neglected, especially when the spectrum is energetically averaged in one-third octave bands, and FFT analysis has not been done.

Shifting of the frequency spectrum due to the Doppler effect should be considered when an UAV is flying at high speeds. Since the source is moving, the centre of each new wavefront is slightly displaced to the right. As a result, an observer in front of the source will hear a higher frequency, and an observer behind the source will hear a lower frequency.

The background noise levels were measured in 1-min intervals. Five independent 1-min measurements were conducted just before and after the flying procedures, mainly with the purpose to correct the measured equivalent noise level of UAV noise. The background noise level was measured during intervals when there were no flight operations or between events when UAV was far away from the measurement location. Whenever any noisy event close to the measurement site was observed, the measurement was stopped and repeated. If the background noise levels (L_res_) were known, the measured total sound pressure levels (L′) could be corrected to obtain only the noise level produced by drone activities (L). If the difference between L_A_,_eq_ (t) and the assumed background noise level in that interval was known, the correction could be made using Equation (1) [46]:(1)$$L=10\cdot \log_{10}\left(10^{\frac{L^{\prime }}{10}}-10^{\frac{L_{\mathrm{res}}}{10}}\right)=10\cdot \log_{10}\left(10^{\frac{L^{\prime }}{10}}\left(1-10^{\frac{L_{\mathrm{res}}-L^{\prime }}{10}}\right)\right)\;L=L^{\prime }+10\cdot \log_{10}\left(1-10^{-0.1\cdot \left(L^{\prime }-L_{\mathrm{res}}\right)}\right)$$

If the difference between specific and background noise was larger than 10 dBA, then the correction is negligible.

## 3. Results and Discussion

### 3.1. UAV Noise Analysis

The sound pressure waveforms were recorded at the microphone position, and the integration procedure was used to find environmental noise parameters: fast-time integration constant and frequency A-weighted equivalent sound pressure levels in 1 s intervals (L_AF,1s_) during operation procedures, sound exposure level of considered events (L_AE_), A-weighted maximum sound pressure level (L_A,max_), C-weighted peak sound pressure level (L_C,peak_), spectrogram with narrow band frequency analysis, and one-third octave band spectrum during hovering operation according to relevant standards [45,46]. The main parameter observed in this research was the A-weighted sound pressure level (L_AF,1s_) during the considered sound events. As an additional parameter, L_C,peak_ was used to characterise the peak sound pressure levels without weighting to show the presence of low-frequency noise. With the increase of UAV operations and flying over a larger population, this parameter might become more relevant due to possible complaints from the annoyed population. A-weighting spectrum can be used to estimate tonal adjustment, which has to be added to the measured equivalent noise level or sound exposure level if any tones are present in the spectrum, according to the ISO 1996-2:2017 standard. The frequency analysis in one-third octave bands was used to find dominant frequency components during the hovering operation at 4.5 m distance from the microphone.

The GPS coordinates of microphones and drones during the flying procedures were determined with 5 cm precision. The precise determination of the location can be used to estimate the sound power level from the known sound pressure level of the UAV at a known distance and flying parameters. Since the sound power level from the drones has not been determined in this paper, this does not have any influence on the measurement results. Nonetheless, it could be used to estimate environmental noise parameters on other sites in some future research.

The sound pressure levels in integration intervals of 1 s with A-frequency weighting and fast-time integration constant L_AF_(t) were calculated from the recorded sound pressure signals in the time domain.

The measured L_AF_(t) signals for the hexarotor and the quadrotor flying up and down 4 times up to 10 m are shown in Figure 3. The fluctuations around 65 dBA in instantaneous sound pressure level are due to the abrupt change of the rotation speed for preserving stability when the UAV is flying up and down, especially close to the measurement point, when v = 0 (changing direction). The fluctuations are more expressed when the drone is flying down than when flying up. This is averaged by observing several individual events.

The results of L_A,F_(t) signals for hexarotor and quadrotor flying-over procedures at 10 m are shown in Figure 4.

The equivalent sound pressure level was determined for all individual events of interest (flying up, flying down, flying over). The correction due to background noise was implemented on the measured overall equivalent sound pressure level, and the average time of events was determined. Furthermore, sound exposure level (L_AE_ or SEL) was calculated for each sound event by knowing the number of events and corrected equivalent sound pressure level of all considered events.

Noise spectrums for both drones were found from the recorded sound pressure while drones were hovering at height of 1.7 m and at a distance of 4.5 m from the measuring microphone. The frequency content depends on the rotation speed and the number of blades. In this research, the presence of low-frequency noise was detected because of the average rotation speed between 4000 and 5000 rpm and the two-blade propeller of both drones during the hovering procedure.

For better visualisation of each frequency component of a hovering UAV, the spectrogram of the hovering hexarotor is shown in Figure 5. Furthermore, Figure 6 shows the narrow band spectrum averaged in each one-third octave band for both UAVs. The basic frequency of drones was detected in one-third octave band with the central frequency of 125 Hz for the hexarotor and 160 Hz for the quadrotor, as shown in Figure 6. There was also a second harmonic of the fundamental frequency component visible in one-third octave band with 250 Hz centre frequency for the hexarotor. It should be noted that the noise spectrum was different during the flight as compared to drone hovering, due to different rotational speeds of engines.

There is a vast number of harmonics appearing in the FFT spectrum. However, by using one-third octave band analysis, these components are energetically averaged with aerodynamic noise. This may be the reason why higher tonal components at 5 kHz found in another research [27] are not that visible in Figure 6 as in FFT spectrum. The one-third octave band is used here because the tonal adjustment according to the ISO 1996-2:2017 is determined in that way and not from the FFT spectrum (ANNEX K in ISO 1996-2:2017). For detailed analysis of the frequency components of different drones, interested readers may refer to [30].

The frequency spectrum is shown only for the point when the UAV was hovering at constant height. Even though the spectrum can be presented in 1 s intervals for each height, the main idea behind analysing the spectrum in this research is only to determine whether there was any tonal component, how much it stood out from the adjacent components, and if the tonal adjustment had to be added to the SEL or L_A,eq_, according to the ISO 1996-2:2017 (ANNEX K). This would be the case if drone noise has to be assessed according to the national legislation.

Furthermore, L_C,peak_ and L_A,peak_ parameters were found from the recorded sound pressure signals for each individual event, and their analysis was conducted in MATLAB. The averaged values were obtained from several considered events, and the results are shown in Table 3.

By knowing the A-weighted sound pressure level in 1 s intervals L_A,F,1s_ (t), the A-weighted sound exposure level L_AE_ was estimated during the considered procedures (by averaging of independent events during flying up and down and flying-over procedures). The averaged results for both drones and for each operation procedure are given in Table 3.

The experimental measurement uncertainty was found for each parameter by repeating the measurements for each considered event which was performed (flying up and down, flying over) under the same conditions (same operating conditions, same measurement equipment). The average values for A-weighted equivalent levels and SEL were found using the energetical averaging with the most weighting on the larger value, while for L_A,max_, the arithmetic averaging as proposed in the ISO 1996-2:2017 standard was used. In addition, standard deviation was calculated for each parameter and experimental measurement uncertainty. Detailed procedure for obtaining the combined measurement uncertainty for L_A,eq_ is explained in ISO 1996-2:2017, which takes into account the measurement uncertainty of the instrument (u_ins_), the influence of background noise (u_back_), and meteorological conditions (u_met_). For more information about measurement uncertainty determination, interested readers can refer to [47]. The results for experimental measurement uncertainty due to source activity (u_sou_) for each considered parameter are given in Table 4.

Two pivotal observations from the UAV noise analysis will be emphasised herein. The first one is that measured parameters (L_AE_, L_C,peak_, L_A,peak_) at the same measurement point of the hexarotor are higher than that of the quadrotor in the flying up and down and flying-over procedures. The second observation is that L_C,peak_ is higher than L_A,peak_ due to the presence of low-frequency noise for both drones. In the hovering procedure, there are visible tonal components (in 125 Hz, one-third octave band for hexarotor and in 160 Hz, one-third octave band for quadrotor). The test for prominent tones in the spectrum was done according to ISO 1996-2:2017 (Annex K), and it is visible that the tone in one-third octave band with centre frequency 160 Hz is more than 8 dB higher than the average levels in the adjacent one-third octave bands (125 Hz and 200 Hz). This means that tonal adjustment K_T_ should be added to the measured equivalent sound pressure level to get the rating sound pressure level when this procedure is considered. Nevertheless, a more detailed tonal analysis during different flying procedures is needed.

Although the main focus of this research has been placed on the assessment of UAV noise impact on the population in general, an additional analysis was conducted to determine the 8-h time-weighted average (TWA) noise levels to which persons directly operating the drones are exposed during working hours. Namely, there are different occupational regulations and standards established to protect workers against the health effects of noise exposure when certain values (or limits) are reached. For instance, the Recommended Exposure Limit (REL) established by the National Institute for Occupational Safety and Health (NIOSH) for noise is 85 decibels, using the A-weighting frequency response over an 8 h average [48]. In addition, there are legal limits, or the Permissible Exposure Limit (PEL), set to 90 dBA by the Occupational Safety and Health Administration (OSHA) for all workers for an 8-h noise exposure in the workplace [48]. Furthermore, maximum allowable daily noise dose, expressed in percentages, is also defined in the occupational standards, and it takes into account both the sound exposure level and the duration. Considering the A-weighted sound exposure levels (LAE) presented in Table 3, it can be concluded that only hexarotor operations during flying up and down procedures exceed the Recommended Exposure Limit, while the noise levels for both drones during all other operations are below the Permissible Exposure Limit. The maximum amount of allowable exposure time after which the noise level of hexarotor operations during ascent and descent becomes hazardous is 5 h and 40 min, or 70.8% of the 8 h working time. Having in mind the current usage of drones, it is unlikely that the maximum allowable daily noise dose will be exceeded. Nevertheless, further analysis might be needed to investigate such claims.

### 3.2. Survey Results Analysis

The responses from 31 survey participant were collected and analysed, and the results are presented herein. Respondents’ answers were first analysed using descriptive statistics. Furthermore, the Shapiro–Wilk test of normality was used to check whether the sets of answers exhibited concurrence with normal distribution, while Wilcoxon matched-pair singed-rank test was used to compare the medians of the two UAV types used.

When conducting a survey, the questionnaire used as the measurement instrument to collect data for the analysis can introduce a bias. In order to examine if a bias was present in this study, Harman’s one-factor test for common method bias was performed. The results showed that the total variance extracted by one factor was 44.547%, which was less than the recommended threshold of 50%, meaning that no bias was introduced.

A box and whisker plot were used to summarise the responses measured on a continuous scale from −5 (completely disagree) to 5 (completely agree). Figure 7 shows the extent to which respondents agreed or disagreed with eight statements related to the operations of the two UAVs.

For all statements, except for the third one, medians for both UAV types are located on the same side of the scale. This implies that more than half of the respondents shared the same attitude towards both drones. These attitudes are positive for statements regarding the negative impact on the mood (S1), safety issues due to possibility of UAV crash (S2), and difficulties in communication (S7), while they are negative for statements regarding the negative impact on focus even when performing simple mental tasks (S4), permanent hearing loss (S5), ringing in ears due to short-term exposure (S6), and disapproval of UAV operations in people’s vicinity (S8). With regard to the third statement, most of the participants disagree that daily quadrotor operations would have a negative impact on their quality of life, while the overall opinion for the hexarotor operations is the opposite.

While most of the box plots are comparatively tall, suggesting that participants hold quite versatile opinions about the presented statements, for several of them (such as statements S1 for the hexarotor and S6 for the quadrotor), comparatively short box plots indicate that the respondents have a high level of agreement with each other.

Even though the differences between the medians among the two drones are obvious and visible (medians are more affirmative for the hexarotor for all statements), additional analysis was conducted to examine whether there was a statistically significant difference between the medians of the two UAV types used.

Before comparing the medians, it is pivotal to check whether the data is normally distributed to be able to use appropriate tests (parametric or non-parametric). Table 5 shows the results of Shapiro–Wilk test of normality chosen for this purpose.

The Shapiro–Wilk test of normality showed that only two sets of answers (Statements 3 and 4 for the hexarotor) were normally distributed. The Shapiro–Wilk test was chosen because it was proven to have the best power for a given significance level [49]. This was confirmed by examining the Q-Q plots as well (omitted for brevity). Thus, the decision was made to compare the medians by using the non-parametric tests for related samples, namely the Wilcoxon matched-pair singed-rank test. The tests were performed on the pairs of the same statements regarding different UAVs (quadrotor vs. hexarotor). The results are shown in Table 6.

The null hypothesis was that the differences of medians between the two UAV types equalled zero. For statement 1, the Wilcoxon signed-rank test revealed that the scores representing the noise perception of the quadrotor were significantly lower (*Md* = 0.60, *n* = 31) compared to the those of the hexarotor (*Md* = 1.70, *n* = 31), *z* = −4.20, *p* < 0.001, with *r* = 0.53. For all the statements, except for the statement S8, the tests showed that the responses were significantly different between the UAV types, and thus the null hypothesis was rejected. This shows that the participants were able to recognise the difference between these two UAVs and had a generally more negative experience with the hexarotor.

After expressing their attitudes towards eight statements for the two drones, the participants were asked to state their opinion about the two most important reasons (out of the five given) for not introducing drone operations into everyday life. Figure 8 shows the percentage of respondents who chose suggested concerns regarding the introduction of UAVs. Most of the respondents (69% of them) were concerned about possible injuries due to possible accidents caused by drone operations, while only 3% of the participants were afraid that drone noise could permanently damage people’s hearing.

As opposed to the previous question, participants were also asked to choose two most important reasons for introducing drone operations into everyday life. Results are presented in Figure 9. As the main reason for UAV introduction, 88% of the respondents recognised the ability of UAV technology to improve the quality of life. The least desirable reason why respondents would like to use UAVs is for urban air mobility (taxi operations).

Lastly, given the overall experience in this research, the respondents were asked to rate whether the benefits of introducing drone operations into everyday life outweighed their negative impact. Figure 10 shows the results in the same way as Figure 7 and gives an overview of the overall opinion of the respondents about the benefits and drawbacks of UAV introduction. More than 80% of the respondents believed that there are more pros than cons for introduction of UAVs into everyday life.

## 4. Conclusions

Over time, the use of drones will undoubtedly increase. One of the major challenges to be overcome will be the noise exposure and human safety. In this paper, noise measurements of a quadrotor and a hexarotor in flying up and down, hovering, and overflight procedures have been reported. Respondents of good hearing confirmed by audiogram measurement, who had participated in the survey during the UAV noise measurement, gave their subjective assessments on UAV noise perception. UAV noise measurements and survey results were analysed and related.

Two important conclusions can be drawn from the UAV noise analysis. The first one is that the measured parameters (L_AE_, L_C,peak_, L_A,peak_) at the same measurement point for the hexarotor were higher in the flying up and down and flying-over procedures compared to those for the quadrotor. The second conclusion is that L_C,peak_ was higher than L_Apeak_ due to the presence of low frequency noise for both drones. Tonal components were visible in the hovering procedure (in 125 Hz, one-third octave band for the hexarotor and in 160 Hz, one-third octave band for the quadrotor), which is in line with the number of propeller blades and approximate RPM for each drone.

Moreover, a few interesting conclusions can be made from the survey result analysis. The participants were able to recognise the difference between quadrotor and hexarotor noise and had a generally more negative experience with the hexarotor. This subjective assessment corresponded to the objective difference in the noise levels between the two UAVs, which was also confirmed by UAV noise measurements. Furthermore, 69% of the respondents were concerned about the possible injuries due to possible accidents caused by drone operations, while only 3% of the participants were afraid that drone noise could permanently damage people’s hearing. This means that more work is needed on the safety of drone operations. As the main reason for UAV introduction into everyday life, 88% of the respondents recognised the ability of UAV technology to improve the quality of life. The least desirable reason why respondents would like to use UAVs was for taxi operations, which is most likely due to doubts about the safety of drone operations in general. And finally, more than 80% of the respondents believed that there are more pros than cons for introduction of UAVs into everyday life.

Given the comprehensive results, this research concludes that there is a positive attitude towards the introduction of more drone operations in the future. Assessment of UAV noise perception can be further expanded by noise analysis of rotocopters with a larger number of engines (such as an octocopter), rotocopters with a higher payload, and generally of different performance. Furthermore, a more detailed tonal analysis during different flying procedures will be conducted as part of future research. Although the preliminary analysis on the recommended 8-h time-weighted average doses for subjects directly operating with drones during longer noise-exposure periods did not indicate any exceedance of permissible exposure limits, this topic should be investigated more thoroughly in future studies.

A comprehensive study involving simultaneous operations of several drones with more survey participants represents one of the possible advancements that will certainly be considered in future research efforts.

## Figures and Tables

**Figure 1 ijerph-18-06202-f001:**
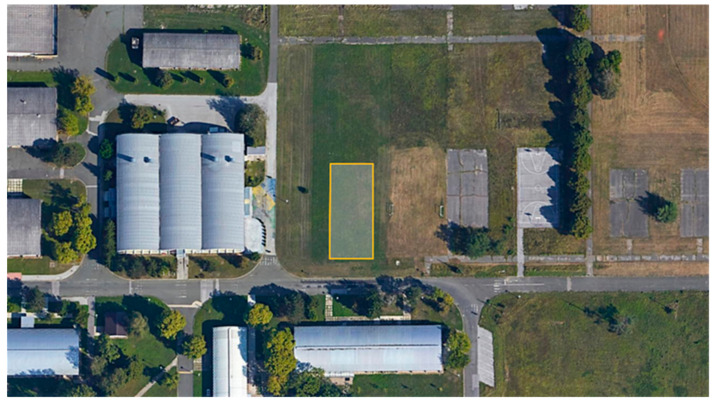
The measurement polygon.

**Figure 2 ijerph-18-06202-f002:**
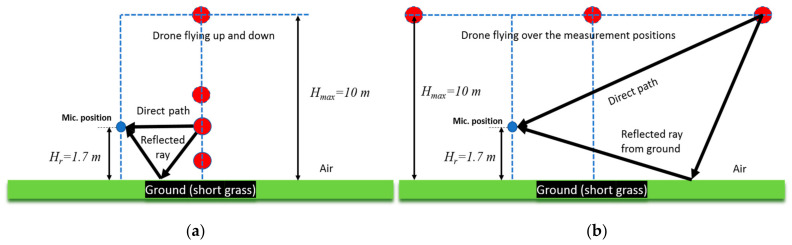
The measurement situations with microphone position and initial flying point for the two procedures: (**a**) flying up and down; (**b**) flying over.

**Figure 3 ijerph-18-06202-f003:**
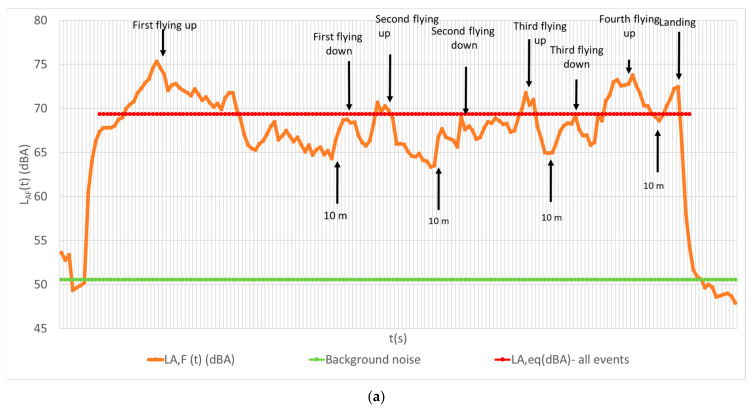
The measurement signals for flying up and down procedures for: (**a**) hexarotor; (**b**) quadrotor.

**Figure 4 ijerph-18-06202-f004:**
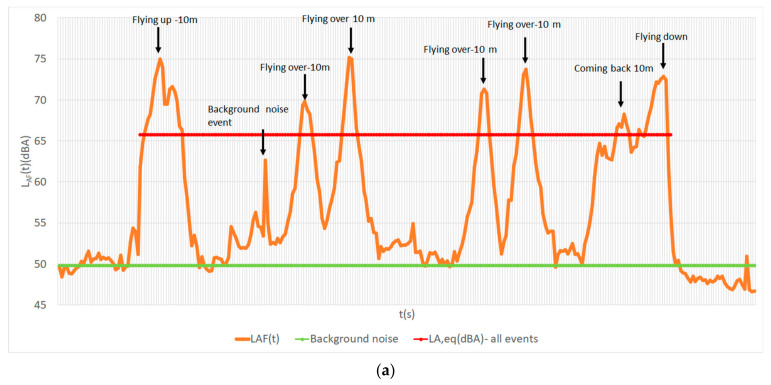
The measurement signals for flying-over procedures for: (**a**) hexarotor; (**b**) quadrotor.

**Figure 5 ijerph-18-06202-f005:**
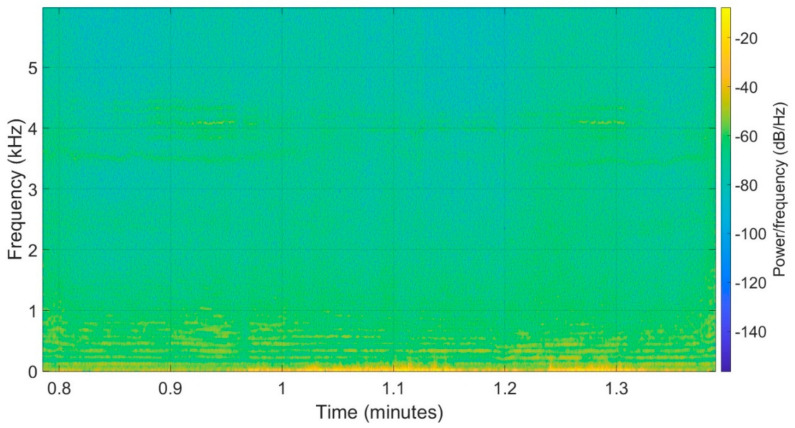
Spectrogram at the time when hexarotor is flying down (0.85–1 min), hovering (1–1.2 min), and flying up (1.2–1.4 min) closest to the microphone position.

**Figure 6 ijerph-18-06202-f006:**
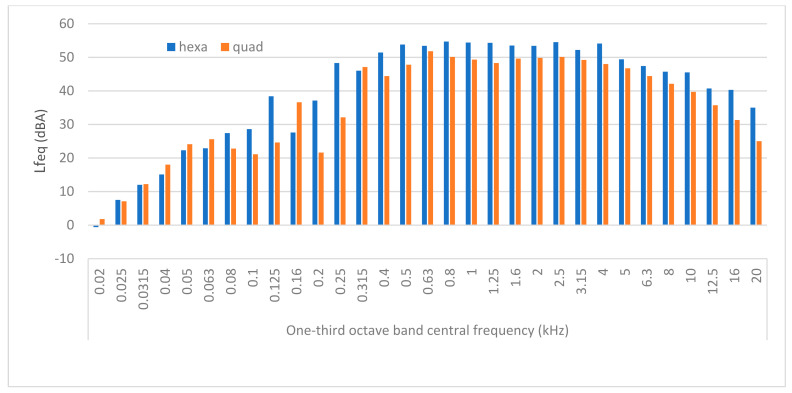
One-third octave band spectrums for both drones during the hovering procedure.

**Figure 7 ijerph-18-06202-f007:**
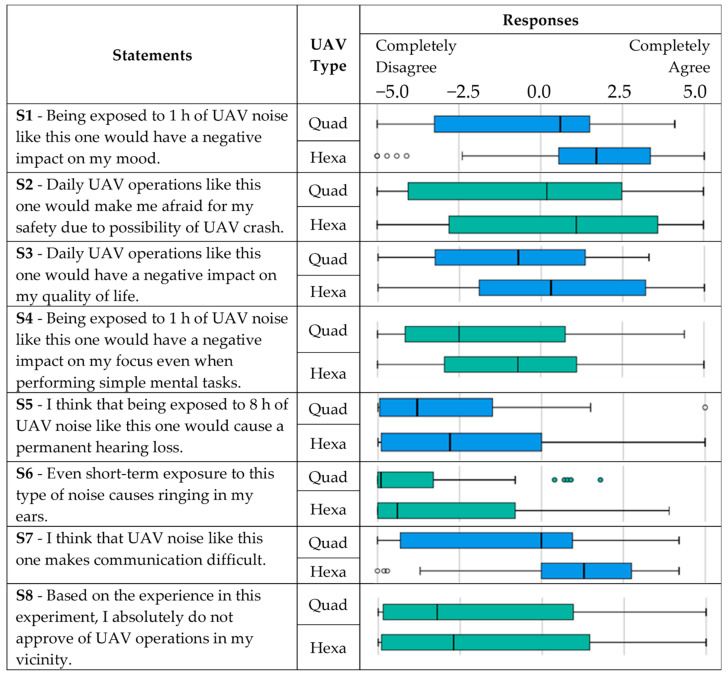
Survey responses to statements for the two UAVs.

**Figure 8 ijerph-18-06202-f008:**
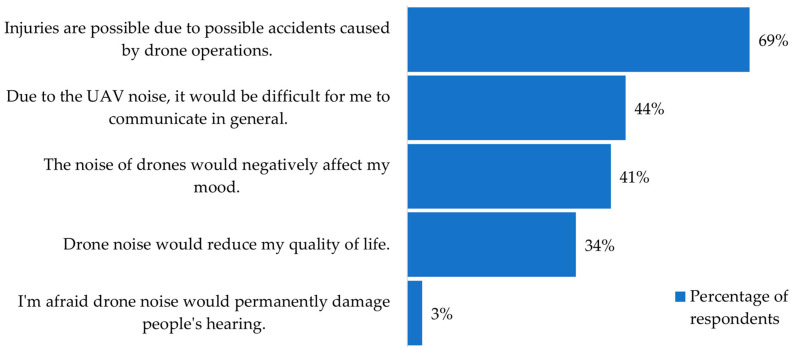
Concerns regarding introduction of UAVs.

**Figure 9 ijerph-18-06202-f009:**
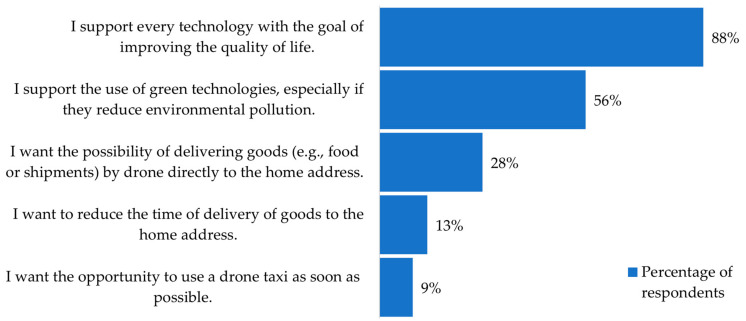
Opportunities regarding introduction of UAVs.

**Figure 10 ijerph-18-06202-f010:**
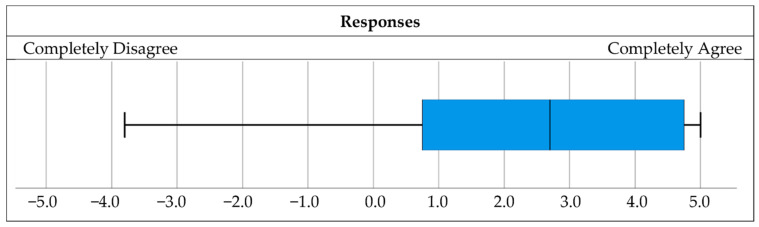
The benefits vs. negative impacts of UAVs.

**Table 1 ijerph-18-06202-t001:** Meteorological conditions during UAV noise measurements and respondents’ assessments.

Measurement Day/Parameter	Temperature (°C)	Relative Humidity (%)	Wind Speed Perpendicular	Wind Speed Parallel	Cloudiness (0–8) According to ISO 1996-2:2017
			on Microphone Membrane Surface (m/s)		
1st day	7.8	68	1.2	0.8	8
2nd day	6.5	59	0.7	0.4	8

**Table 2 ijerph-18-06202-t002:** UAV characteristics.

	Small Commercial Quadrotor	Custom-Built Hexarotor
Number of motors	4	6
Weight (g)	2845	5660
Dimensions w/o propellers (mm)	438 × 451 × 301	960 × 960 × 530
Motor velocity constant (KV)	420	340
Nominal voltage (V)	22.2	22.2
Propeller size and pitch	13″ × 4.5″	17″ × 5.8″
Number of propeller blades	2	2
Approximate RPM in hover	5000	4000

**Table 3 ijerph-18-06202-t003:** Overall measurement results.

	L_AE_ (dBA)		L_A,max_ (dBA)		L_A,peak_ (dBA)		L_C,peak_ (dBC)	
	Flying up-down	Flying over	Flying up-down	Flying over	Flying up-down	Flying over	Flying up-down	Flying over
Hexarotor	86.5	78.8	75.3	72.5	95.3	90.0	97.5	91.8
Quadrotor	81.0	70.9	70.2	65.4	88.3	81.5	91.0	83.0

**Table 4 ijerph-18-06202-t004:** Experimental measurement uncertainty (u_sou_).

	L_AE_ (dBA)		L_A,max_ (dBA)		L_A,peak_ (dBA)		L_C,peak_ (dBC)	
	Flying up-down	Flying over	Flying up-down	Flying over	Flying up-down	Flying over	Flying up-down	Flying over
Hexarotor	1.0	0.8	0.5	1.1	1.9	1.9	2.0	1.6
Quadrotor	1.5	1.1	0.6	1.2	1.0	0.9	1.1	0.6

**Table 5 ijerph-18-06202-t005:** Shapiro–Wilk test of normality.

Statements	Quadrotor		Hexarotor	
	Statistic	Sig.	Statistic	Sig.
S1	0.908	0.012	0.883	0.003
S2	0.896	0.006	0.898	0.007
S3	0.921	0.024	0.934	0.057
S4	0.919	0.023	0.961	0.301
S5	0.836	0.000	0.874	0.002
S6	0.665	0.000	0.799	0.000
S7	0.894	0.005	0.878	0.002
S8	0.839	0.000	0.853	0.001

**Table 6 ijerph-18-06202-t006:** Related samples Wilcoxon signed-rank test summary (quadrotor vs. hexarotor).

Statements	UAV Type	Median	Wilcoxon Signed-Rank Test Statistic	Asymptotic Sig. (2-Tailed)
S1	Quadrotor	0.6	−4.2000	<0.001
	Hexarotor	1.7		
S2	Quadrotor	0.2	−2.6940	0.007
	Hexarotor	1.1		
S3	Quadrotor	−0.7	−3.6220	<0.001
	Hexarotor	0.3		
S4	Quadrotor	−2.5	−3.1510	0.002
	Hexarotor	−0.7		
S5	Quadrotor	−3.8	−1.9660	0.049
	Hexarotor	−2.8		
S6	Quadrotor	−4.9	−2.1130	0.035
	Hexarotor	−4.4		
S7	Quadrotor	0.0	−3.6800	<0.001
	Hexarotor	1.3		
S8	Quadrotor	−3.2	−1.7560	0.079
	Hexarotor	−2.7		

## Data Availability

Not applicable.

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
