# Peer review of "Comparative UAV Noise-Impact Assessments through Survey and Noise Measurements"

_ijerph, 2021, doi:10.3390/ijerph18126202_

Round 1

Reviewer 1 Report

1. Brand names should be generally avoided due to no-ad policy in technical/scientific papers: LabView, NI cDAQ 9171, Nor140, Bell audiometer, Grass microphone, Testo sensors, etc. Consider to use brief specs instead, or instrument grade where applicable ("Class 1 sound analyzer" instead "Nor140" would be adequate, for instance)

Reviewer 2 Report

Dear authors,

please find enclosed the referee report on the nice paper "Comparative UAV noise impact assessments through survey and noise measurements", submitted for publication in International Journal of Environmental Research and Public Health.

Kind regards.

Reviewer 3 Report

Improve reference there are same paper about the noise emitted by UAVs Iannace et al, Papa at al., Ciaburro et al.   Fig. 5 shows the data in dBA, it is not necessary to report below the 100 Hz value.   In some acoustic measurements performed by different authors at 5 kHz a tonal component was found I can't see it in Fig. 5. Can you explain why?   Insert more figures of the frequency spectrum also in dB lin.   You could also insert a Spectrogram with different scenarios.   Explain how you did the listening tests, it is not clear how the listening tests were performed. The noise changes according to the flight attitude, how did you evaluate this configuration?   Measurements are performed outdoors. But how long can a UAV fly? Not much! has this been considered?   you could implement the test questions on how they perceive noise because the tonal component is present, which I don't see here.

Reviewer 4 Report

While the topic of the work is interesting and relatively novel, the paper and its experimental design do not seem to be particularly well-prepared. Measurements claim to be referring to standards, but I struggle to follow the protocols.

For the subjective part of the experiment (questionnaire), I think I read the paper twice, and I still can't tell for sure how the questionnaires/interviews took place.

I am skeptical about the subjective responses, the questions are posed in a very negative way, which certainly introduced a strong bias for the respondents.

Statistics are not reported in standardized way.

Round 2

Reviewer 3 Report

no

Reviewer 4 Report

Thanks for your comments, I respond to points that I have think have been overlooked in my review:

Response 1: The authors greatly appreciate the comment given by the Reviewer 4. However, it would be more beneficial if the reviewer specified which protocols and standards were not followed. In this way, the authors can neither improve the manuscript nor comply to the comments provided.

There is plenty of protocols to follow, starting from the environmental noise measurements ISO (e.g., ISO 1996-2:2017, adapted for the specific source, I don't think what the authors measured is compliant), as well as characterizing the actual sound power emitted by the drones  (see for instance: ISO 3740:2019 and related standards), which I think should have been in the first place, and discussed.

Response 3: While creating the survey questions, the authors consulted an expert in statistics. While the questions were generally posed in a negative way, it did not introduce any bias for the respondents. The overall results of the survey confirm this.

I don't see how this response is addressing my point. An expert in statistics may have offered advice on the analysis and or psychometric aspects, like scales, etc. - but not on the actual "phrasing" of the question. How can you claim that no bias was introduced? How do you test it?

Response 4: As stated before, although the authors greatly appreciate every reviewer’s comment, the manuscript cannot be improved if the reviewer does not specify what is considered to be the standardised way in statistical reporting. The obtained results were analysed using descriptive statistics, which is the basic way of describing research findings. Furthermore, Shapiro-Wilk test of normality is commonly used to check whether data are normally distributed. The same goes for Wilcoxon matched-pair singed-rank test and box and whisker plot.

I did not specify as I felt it was common practice. However, I can refer you to the guidance for reporting results of Common Statistical Tests in APA Format. Ask the "expert in statistics", they should know.
